# On Properties of Distance-Based Entropies on Fullerene Graphs

**DOI:** 10.3390/e21050482

**Published:** 2019-05-10

**Authors:** Modjtaba Ghorbani, Matthias Dehmer, Mina Rajabi-Parsa, Abbe Mowshowitz, Frank Emmert-Streib

**Affiliations:** 1Department of Mathematics, Faculty of Science, Shahid Rajaee Teacher Training University, Tehran 6785-136, Iran; 2Steyr School of Management, University of Applied Sciences Upper Austria, 4400 Steyr Campus, Austria; 3Department of Biomedical Computer Science and Mechatronics, UMIT, 6060 Hall in Tyrol, Austria; 4College of Artificial Intelligence, Nankai University, Tianjin 300350, China; 5Department of Computer Science, The City College of New York (CUNY), New York, NY 10031, USA; 6Predictive Medicine and Data Analytics Lab, Department of Signal Processing, Tampere University of Technology, 33720 Tampere, Finland; 7Institute of Biosciences and Medical Technology, 33520 Tampere, Finland

**Keywords:** graph entropy, eccentricity, Hosoya polynomial, 05C70, 05C07, 05C35, 92E10

## Abstract

In this paper, we study several distance-based entropy measures on fullerene graphs. These include the topological information content of a graph $I_{a}\left(G\right)$, a degree-based entropy measure, the eccentric-entropy $If_{\sigma }\left(G\right)$, the Hosoya entropy H(G) and, finally, the radial centric information entropy $H_{ecc}$. We compare these measures on two infinite classes of fullerene graphs denoted by $A_{12n+4}$ and $B_{12n+6}$. We have chosen these measures as they are easily computable and capture meaningful graph properties. To demonstrate the utility of these measures, we investigate the Pearson correlation between them on the fullerene graphs.

## 1. Introduction

Graph entropy measures have been used in applied network sciences to characterize networks, quantitatively [1,2,3]. Such measures were first introduced in the 1950s in studies of biological and chemical systems. Seminal work in this area was done by Rashevsky [4] and Mowshowitz [3,5,6,7,8], who investigated mathematical properties of entropy measures. IN particular, Mowshowitz [3] interpreted the topological information content of a graph as the entropy of the underlying graph topology. To date, numerous graph entropies have been developed and applied to problems in both theoretical and applied disciplines (see [1,2,3]). Examples include problems in biology, computational biology, mathematical chemistry, web mining, and knowledge engineering concerned with structural properties of networks (see [1,2,3,5,6,7,8,9,10,11,12,13,14,15,16]).

As numerous network measures have been developed so far [3], it is often difficult to choose an appropriate measure for a given class of graphs. This is so for several reasons including the following: (1) The graphs in a given class may be characterized by special structural properties such as symmetry, cyclicity, linearity, and so forth, and not every measure is able to quantify those structural properties in a meaningful way; (2) a particular graph measure relies on a special graph invariant or a combination thereof. For instance, the well-known topological information content $I_{a}$ [3] has been used as a symmetry measure as it is based on the automorphism group of a graph. Yet, $I_{a}$ may not be a good measure for distinguishing graphs on cyclicity or other structural properties. In short, the measure one chooses must be appropriate for the structural feature of interest.

A fullerene is a molecule composed of carbon atoms in the form of many shapes such as a hollow sphere, ellipsoid, tube, etc (see [17]). In the mathematical meaning, a fullerene is a cubic 3-connected planar graph with pentagonal and hexagonal faces. For more details of the mathematical aspects of fullerenes, see [18,19,20,21,22]. In this paper, we compare the topological information content of a graph $I_{a}\left(G\right)$, the eccentric-entropy $If_{\sigma }\left(G\right)$, the Hosoya entropy H(G), the radial centric information entropy $H_{ecc}$, and a special degree-based entropy on two infinite classes of fullerene graphs, namely $A_{12n+4}$ and $B_{12n+6}$. We emphasize that these measures have already been explored extensively and possess a known structural interpretation. In addition, fullerene graphs play an important role in mathematical chemistry and related disciplines. Therefore, we want to investigate the sensitivity of these five measures to the structural properties of fullerenes. This study is intended as an aid in selecting a measure capable of capturing the structural information of fullerenes. On the other hand, graph measures are at least relevant to the fullerene reactivity [20]. Additionally, entropy-based graph measures may relate to non-equilibrium physicochemical processes (see [23,24]). As for fullerenes, there are direct applications of information entropy to rationalizing the processes of fullerene [25] and endofullerene [26] formation.

## 2. Concepts and Terminology

All graphs considered in this paper are simple, connected, and finite. Let *x* and *y* be two arbitrary vertices of graph *G*. The distance between *x* and *y* is the length of the shortest path connecting them, denoted by d(x,y).

Let Γ be a group and Ω be a non-empty set. An action of group Γ on set Ω is a function φ:Γ×Ω→Ω, where (g,α)→φ(g,α), that satisfies the following two properties (we denote φ(g,α) as $\alpha^{g}$): $\alpha^{e}=\alpha$ for all α in Ω and ${\left(\alpha^{g}\right)}^{h}=\alpha^{gh}$ for all g,h in Γ. The orbit of an element α∈Ω is denoted by $\alpha^{\Gamma }$, and it is defined as the set of all $\alpha^{g},\;g\in \Gamma$.

Let X=(V,E) be a graph with vertex set *V* and edge set *E*. A bijection *f* on *V* that preserves edge set *E* is called an automorphism of *X*. In other words, the bijection *f* on V(X) is an automorphism if and only if f(u)f(v) is an edge in *E* (the image of vertex *u* is denoted by f(u)) whenever e=uv is an edge in *E*. The set of all automorphisms of *X*, denoted by Aut(X), forms a group under the composition of mappings. This group acts transitively on the set of vertices if for any pair of vertices *u* and *v* in V(X), there is an automorphism g∈Aut(X) such that g(u)=v. In this case, we say that *X* is vertex-transitive. An edge-transitive graph can be defined similarly.

The stabilizer of a vertex *v* under the action of A=Aut(G) is the set of automorphisms that fix *v* and is denoted by $A_{v}$. A group-theoretic result of special importance regarding the proofs in Section 4 is the orbit-stabilizer theorem, which states that $|v^{A}\left|\right|A_{v}\left|=|A\right|$.

## 3. Entropy of Graphs

The general Shannon entropy [27] is defined by $I\left(p\right)=-\sum_{i=1}^{n}p_{i}\log\left(p_{i}\right)$ for finite probability vector *p*. Let $\lambda =\sum_{j=1}^{n}\lambda_{j}$ and $p_{i}=\lambda_{i}/\lambda ,\left(i=1,2,\ldots ,n\right).$ Generally, the entropy of an *n*-tuple $\left(\lambda_{1},\lambda_{2},\ldots ,\lambda_{n}\right)$ of real numbers is given by
(1)$$I\left(\lambda_{1},\lambda_{2},\ldots ,\lambda_{n}\right)=-\sum_{i=1}^{n}p_{i}\log\left(p_{i}\right)=\log\left(\sum_{i=1}^{n}\lambda_{i}\right)-\sum_{i=1}^{n}\frac{\lambda_{i}}{\sum_{j=1}^{n}\lambda_{j}}\log\lambda_{i}.$$

There are many different ways to associate an n-tuple $\left(\lambda_{1},\lambda_{2},\ldots ,\lambda_{n}\right)$ to a graph *G* (see [1,2,8,10,11,12,13,14,15,16,28]). A classical graph entropy measure, namely the topological information content due to Mowshowitz [3], is defined by
(2)$$I_{a}\left(G\right)=-\sum_{i=1}^{k}\frac{\left|N_{i}\right|}{\left|V\right|}\log\left(\frac{\left|N_{i}\right|}{\left|V\right|}\right),$$
where $N_{i}$ (1 ≤ *i* ≤ *k*) is a set of similar vertices (which means they are in the same orbit). The collection of *k* orbits $N_{1},\ldots ,N_{k}$ defines a finite probability scheme in an obvious way. It is well-known that $I_{a}$(*G*) reaches its maximum value for an identity graph, i.e., one whose automorphism group consists of the identity alone [3].

Entropy measures have been applied to networks/graphs extensively (see, e.g., [1,2,3]). There are many so-called information functionals that can be used to characterize a graph by means of an entropy measure defined by Equation (Equation 3). Because vertex eccentricity has meaningful properties (see [28,29]), we will apply this measure in our analysis together with other graph entropies. The eccentricity of vertex *v* is $\sigma \left(v\right)=max_{u\in V}d\left(u,v\right)$, where d(u,v) is the distance between vertices *u* and *v*. For a vertex $v_{i}\in V$, we define *f* as $f\left(v_{i}\right):=c_{i}\sigma \left(v_{i}\right)$ where $c_{i}>0$ for 1≤i≤n (see [3]). The entropy based on *f* denoted by $If_{\sigma }\left(G\right)$ is defined as follows:(3)$$\begin{array}{c} If_{\sigma }\left(G\right)=\log\left(\sum_{i=1}^{n}c_{i}\sigma \left(v_{i}\right)\right)-\sum_{i=1}^{n}\frac{c_{i}\sigma \left(v_{i}\right)}{\sum_{j=1}^{n}c_{j}\sigma \left(v_{j}\right)}\log\left(c_{i}\sigma \left(v_{i}\right)\right). \end{array}$$

If $c_{i}^{\prime }s$ are equal, then
(4)$$\begin{array}{c} If_{\sigma }\left(G\right)=\log\left(\sum_{i=1}^{n}\sigma \left(v_{i}\right)\right)-\sum_{i=1}^{n}\frac{\sigma (v_{i})}{\sum_{j=1}^{n}\sigma \left(v_{j}\right)}\log\left(\sigma \left(v_{i}\right)\right). \end{array}$$

For further information about existing graph entropy measures, see [3,10,30,31,32,33,34].

In addition, we apply a special degree-based entropy D(G) defined by [3]
(5)$$\begin{array}{c} D\left(G\right)=\log\left(\sum_{i=1}^{n}c_{i}\deg\left(v_{i}\right)\right)-\sum_{i=1}^{n}\frac{c_{i}\deg\left(v_{i}\right)}{\sum_{j=1}^{n}c_{j}\deg\left(v_{j}\right)}\log\left(c_{i}\deg\left(v_{i}\right)\right). \end{array}$$

It is evident that other degree-based entropies can be defined as well (e.g., see [35]). If $c_{i}^{,}s$ are equal, since $\sum_{i=1}^{n}\deg\left(v_{i}\right)=2m$, where *m* is the number of edges, we obtain
(6)$$\begin{array}{c} D\left(G\right)=\log\left(2m\right)-\frac{1}{2m}\sum_{i=1}^{n}\deg\left(v_{i}\right)\log\left(\deg\left(v_{i}\right)\right). \end{array}$$

Given a graph *G* and a vertex u∈V(G), let $\Gamma_{i}\left(u\right)$ be the number of vertices at distance *i*from *u*. Two vertices *u* and *v* are said to be Hosoya-equivalent or *H*-equivalent [8] if $\Gamma_{i}\left(u\right)=\Gamma_{i}\left(v\right)$ for 1 ≤*i*≤*d*(*G*). The family of sets of *H*-equivalent vertices constitutes a partition of the vertices. Let *h* be the number of sets of *H*-equivalent vertices in *G*. The Hosoya entropy (or *H*-entropy) of *G* (introduced in [8]) is given by
(7)$$H\left(G\right)=-\sum_{i=1}^{h}\frac{|X_{i}|}{\left|V\right|}\log\left(\frac{|X_{i}|}{\left|V\right|}\right).$$

Another entropy measure we use here relates to vertex eccentricity. The eccentric-entropy of graph *G* denoted by $C_{ec}\left(G\right)$ is defined by the number of different eccentricities of vertices [35]. Let $C_{ec}\left(G\right)=k$ and $Y_{i_{1}},\cdots ,Y_{i_{k}}$ be the sets of the different eccentricities. For instance, $Y_{i_{j}}$, 1≤j≤k is the set of all vertices with eccentricity equal to $i_{j}$. Then, the radial centric information entropy (or simply radial entropy) is defined by [36]
(8)$$\begin{array}{c} H_{ecc}=-\sum_{i=1}^{k}\frac{|Y_{i}|}{\left|V\right|}\log\left(\frac{|Y_{i}|}{\left|V\right|}\right). \end{array}$$

The eccentric sequence of a connected graph *G* represents a list of the eccentricities of its vertices in non-decreasing order. Since there are often many vertices having the same eccentricity, we simplify the sequence by listing them as
$$\left\{\sigma {\left(v_{1}\right)}^{m_{1}},\sigma {\left(v_{2}\right)}^{m_{2}},\cdots ,\sigma {\left(v_{k}\right)}^{m_{k}}\right\}.$$
$\sigma (v_{i})$ is the eccentricity of $v_{i}$; $m_{i}$ is the multiplicity of $\sigma (v_{i})$.

## 4. Main Results

In this section, we consider two infinite classes of fullerene graphs $A_{12n+4}$ and $B_{12n+6}$. Group-theoretic methods are used to determine the orbits of their respective automorphism groups, enabling the computation of symmetry-based entropy. Hosoya entropy is also computed using a method [29] for inferring Hosoya partitions.

In addition, we determine the eccentricity sequence of a fullerene graph, and we calculate the radial centric entropy. Eccentricity entropy and degree-based entropy, defined in the previous section, are also computed. Finally, in Section 4, we compare these entropies in relation to properties of the graphs.

**Lemma** **1.**
*[37] If G is a vertex-transitive graph, then for all x,y∈V(G), σ(x)=σ(y), i.e., all the vertices in a vertex-transitive graph have the same eccentricity.*


**Theorem** **1.**
*[38] Let G be a vertex-transitive graph on n vertices and σ(x) denote the eccentricity of vertex x. For all sequences $c_{1}\geq c_{2}\geq \cdots \geq c_{n}$*
(9)$$If_{\sigma }\left(G\right)=\log\left(\sum_{i=1}^{n}c_{i}\right)-\sum_{i=1}^{n}\frac{c_{i}}{\sum_{j=1}^{n}c_{j}}\log\left(c_{i}\right).$$

*If $c_{i}=c_{j}$ for all i≠j, then $If_{\sigma }\left(G\right)=\log\left(n\right)$.*

*Let $x_{1},x_{2}$ be positive integers. It is clear that the inequality*
(10)$${\left(x_{1}+x_{2}\right)}^{\left(x_{1}+x_{2}\right)}>x_{1}^{x_{1}}x_{2}^{x_{2}},$$
*is satisfied. We are aware of the fact that the Hosoya partition is either an orbit or a union of distinct orbits. Thus, using Equation (Equation 10), we conclude that for an arbitrary graph G, we infer*
(11)$$H_{ecc}\left(G\right)\leq H\left(G\right)\leq I_{a}\left(G\right).$$

*It is not difficult to see that the diameter of $A_{12n+4}$ is d=2n+1. Suppose that for 1≤i≤n, $C_{i}$ is the subset of vertices of $A_{12n+4}$ at distance 2i−1 or 2i from the vertex 1, and $C_{n+1}=\left\{12n,\ldots ,12n+4\right\}$. In other words, for 1≤i≤n, $C_{i}=\Gamma_{2i-1}\left(1\right)\cup \Gamma_{2i}\left(1\right)$ and $C_{n+1}=\Gamma_{2n+1}\left(1\right)$, where $\Gamma_{i}\left(u\right)$ is defined in Section 3.*


**Definition** **1.**
*The i-th layer (1≤i≤n) of a fullerene graph $A_{12n+4}$ is the set of vertices contained in $C_{i}$.*


**Theorem** **2.**
*Consider the fullerene graph $A_{12n+4}$, where n≥4. If n is even, then*
(12)$$I_{a}\left(A_{12n+4}\right)=\log\left(12n+4\right)-\frac{1}{12n+4}\left((12n+3)\log3+6n\right).$$

*If n is odd, then*
(13)$$I_{a}\left(A_{12n+4}\right)=\log\left(12n+4\right)-\frac{1}{12n+4}\left((12n+3)\log3+6n+6\right).$$


**Proof.** Consider the labeling of the fullerene graph $A_{12n+4}$ as shown in Figure 1, and set α=(2,5,8)(3,6,9)…(12n+2,12n+4,12n) and β=(2,8)(3,7)…(12n+2,12n).Clearly, $S_{3}≅\langle \alpha ,\beta \rangle \leq A=Aut\left(A_{12n+4}\right)$. On the other hand, $\left|A|=\right|1^{A}\left|\right|A_{1}|$. Since each automorphism that fixes points 1 and 2 must fix {7,11,17,⋯12n−1,12n+2}, $|A_{1}\left|=\right|A_{1,2}\left|\right|2^{A_{1}}|=2\times 3$. Moreover, $1^{A}=\left\{1\right\}$, and thus |A|=6, which implies that $A≅S_{3}.$The vertex 1 constitutes a singleton orbit. The vertices of the first layer of this graph constitute two orbits,
{2,5,8},{3,4,6,7,9,10}.On the other hand, the *i*-th layer (2≤i≤n) consists of three orbits. The vertices of the *i*-th layer of $A_{12n+4}$ that are the same color (in Figure 2) are in the same orbit.If *n* is even, the vertices of the last layer of Figure 1 make up two orbits: the vertices with odd labels (colors) form one orbit, and the other vertices form a second orbit. If *n* is odd, the vertices of the last layer are in the same orbit. Thus, if *n* is even, the fullerene graph $A_{12n+4}$ possesses one orbit of size 1, 2n+1 orbits of size 3, and *n* orbits of size 6. Thus,
$$\begin{array}{ccc} I_{a}\left(A_{12n+4}\right) & = & \frac{1}{12n+4}\log\left(12n+4\right)+\frac{3(2n+1)}{12n+4}\log\left(\frac{12n+4}{3}\right) \end{array}$$
(14)$$\begin{array}{c} +\frac{6n}{12n+4}\log\left(\frac{12n+4}{6}\right) \end{array}$$
(15)$$\begin{array}{c} =\log\left(12n+4\right)-\left(\frac{6n+3}{12n+4}\log3+\frac{6n}{12n+4}\left(1+\log3\right)\right) \end{array}$$
(16)$$\begin{array}{c} =\log\left(12n+4\right)-\frac{1}{12n+4}\left((12n+3)\log3+6n\right). \end{array}$$If *n* is odd, the fullerene graph $A_{12n+4}$ has one orbit of size 1, 2n−1 orbits of size 3, and n+1 orbits of size 6. Hence,
$$\begin{array}{ccc} I_{a}\left(A_{12n+4}\right) & = & \frac{1}{12n+4}\log\left(12n+4\right)+\frac{3(2n-1)}{12n+4}\log\left(\frac{12n+4}{3}\right) \end{array}$$
(17)$$\begin{array}{c} +\frac{6(n+1)}{12n+4}\log\left(\frac{12n+4}{6}\right) \end{array}$$
(18)$$\begin{array}{c} =\log\left(12n+4\right)-\left(\frac{6n-3}{12n+4}\log3+\frac{6(n+1)}{12n+4}\left(1+\log3\right)\right) \end{array}$$
(19)$$\begin{array}{c} =\log\left(12n+4\right)-\frac{1}{12n+4}\left((12n+3)\log3+6n+6\right). \end{array}$$  □

**Theorem** **3.**
*The fullerene graph $A_{12n+4}$ where n≥4 satisfies*
(20)$$H\left(A_{12n+4}\right)=\log\left(12n+4\right)-\frac{1}{12n+4}\left((12n+3)\log3+6(2n-4)\right).$$


**Proof.** Consider the graph shown in Figure 2. Each set of *H*-equivalent vertices in the *i*-th (i=1,2,3,4,5) layer forms a distinct orbit. For i=6,7,⋯,n, the vertices of each layer constitute three orbits labeled by the numbers 1, 2, and 3. In all of them, vertices with labels 2 and 3 compose *H*-equivalent partitions and the vertices with label 1 compose another *H*-equivalent partition. Finally, the vertices of the outer pentagon in $A_{12n+4}$ are also *H*-equivalent.This means that the vertices of fullerene graph $A_{12n+4}$ are partitioned into 2n+6*H*-equivalence classes such that there exists an equivalence class of size 1, nine equivalence classes of size 3, and 2n−4 equivalence classes of size 6. Hence,
$$\begin{array}{ccc} H(A_{12n+4}) & = & \frac{1}{12n+4}\log\left(12n+4\right)+\frac{27}{12n+4}\log\left(\frac{12n+4}{3}\right) \end{array}$$
(21)$$\begin{array}{c} +\frac{6(2n-4)}{12n+4}\log\left(\frac{12n+4}{6}\right) \end{array}$$
(22)$$\begin{array}{c} =\log\left(12n+4\right)-\left(\frac{27}{12n+4}\log3+\frac{6(2n-4)}{12n+4}\left(1+\log3\right)\right) \end{array}$$
(23)$$\begin{array}{c} =\log\left(12n+4\right)-\frac{1}{12n+4}\left(12n-24+(12n+3)\log3\right). \end{array}$$Carbon nanotubes are members of the fullerene family. A carbon nanotube ($T_{z}\left[m,n\right]$) consists of a sheet with *m* rows and *n* columns of hexagons (see Figure 3). Nanotubes can be pictured as sheets of graphite rolled up into a tube, as shown in Figure 4. Combining a nanotube $T_{z}\left[6,n-10\right]$ with two copies of $B_{1}$ and $B_{2}$ (Figure 5 and Figure 6) yields the fullerene graph $A_{12n+4}$ (see Figure 7).  □

The vertices of fullerene graph $A_{12n+4}$ can be partitioned into three subsets of vertices: the vertices of $B_{1}$, $B_{2}$ and the vertices of the nanotube $T_{z}\left[6,n-10\right]$ (see Figure 5, Figure 6, and Figure 8). The blocks of the Hosoya partition and the eccentricities of the vertices of $B_{1}$ and $B_{2}$ are given in Table 1.

Now consider the nanotube $T_{z}\left[6,n-10\right]$ in fullerene graph $A_{12n+4}$. Each layer of this graph has two equivalence classes (see Figure 8). Let $p_{1},\ldots ,p_{n-10}$ be the Hosoya-equivalent vertices of $T_{z}\left[6,n-10\right]$, i.e., the set $p_{i}$ contains the vertices labeled *i*. Then $ecc(p_{i})=2n-i-9$, where for the subset X⊆V(G), ecc(X)=max{ecc(x):x∈X}.

Thus, the eccentricity sequence of fullerene graph $A_{12n+4}$ is
(24)$$\left\{{\left(2n-i\right)}^{12}\left(1\leq i\leq n-1\right),{\left(2n\right)}^{9},{\left(2n+1\right)}^{7}\right\}.$$

**Theorem** **4.**
*The radial entropy of fullerene $A_{12n+4}\left(n\geq 4\right)$ is*
(25)$$\begin{array}{cc} H_{ecc}\left(A_{12n+4}\right) & =\log(12n+4)\\ & -\frac{1}{12n+4}\left(24(n-1)+(12n+6)\log3+7\log7\right). \end{array}$$


**Proof.** From Equation (Equation 24), we obtain
$$\begin{array}{ccc} H_{ecc}\left(A_{12n+4}\right) & = & \frac{12(n-1)}{12n+4}\log\left(\frac{12n+4}{12}\right)+\frac{9}{12n+4}\log\left(\frac{12n+4}{9}\right) \end{array}$$
(26)$$\begin{array}{cc} + & \frac{7}{12n+4}\log\left(\frac{12n+4}{7}\right) \end{array}$$
(27)$$\begin{array}{cc} = & \log\left(12n+4\right)-\frac{1}{12n+4}\left(12(n-1)\log12+9\log9+7\log7\right) \end{array}$$
(28)$$\begin{array}{cc} = & \log\left(12n+4\right)-\frac{1}{12n+4}\left(24(n-1)+(12n+6)\log3+7\log7\right). \end{array}$$  □

**Theorem** **5.**
*If $c_{i}$’s are equal in Equation (Equation 5), the entropy of fullerene $A_{12n+4}\left(n\geq 11\right)$ is given by*
(29)$$\begin{array}{cc} If_{\sigma }\left(A_{12n+4}\right) & =\log(18n^{2}+14n+7)\\ & -\frac{1}{18n^{2}+14n+7}\left((14n+7)\log(2n+1)\right)\\ & +18n\log(2n)+12A), \end{array}$$
*where $A=\sum_{i=1}^{n-1}\left(2n-i\right)\log\left(2n-i\right)$.*


**Proof.** From Table 1, assuming n≥11, it is clear that there are n+1 types of vertices of the fullerene graph $A_{12n+4}$ with distinct eccentricities. From Equation (Equation 24), one can see that there exist 7 vertices with eccentricity 2n+1, 9 vertices with eccentricity 2n, and 12 vertices with eccentricity 2n−i(1≤i≤n−1). From this, we conclude that
(30)$$\begin{array}{cc} If_{\sigma }\left(A_{12n+4}\right) & =\log\left(7\left(2n+1\right)+9\left(2n\right)+12\sum_{i=1}^{n-1}\left(2n-i\right)\right)\\ & -\frac{1}{32n+7+12\sum_{i=1}^{n-1}\left(2n-i\right)}(7\left(2n+1\right)\log\left(2n+1\right)\\ & +9\left(2n\right)\log\left(2n\right)+12A))=\log\left(18n^{2}+14n+7\right)\\ & -\frac{1}{18n^{2}+14n+7}\left((14n+7)\log(2n+1)\right)\\ & +18n\log(2n)+12A. \end{array}$$  □

**Theorem** **6.**
*The degree-based entropy of fullerene graph $A_{12n+4}$ is*
(31)$$D\left(A_{12n+4}\right)=\log\left(36n+12\right)-\frac{(36n+12)\log3}{36n+12}=\log\left(12n+4\right).$$


**Theorem** **7.**
*Let $B_{12n+6}$ be the fullerene graph with n≥6. Then*
(32)$$I_{a}\left(B_{12n+6}\right)=\log\left(12n+6\right)-\frac{10n+5}{6n+3}.$$


**Proof.** Consider the graph $B_{12n+6}$ shown in Figure 9. Clearly, α,β are automorphisms of fullerene graph $B_{12n+6}$:
$$\begin{array}{c} \alpha =(1,5)(2,4)(6,8)\ldots (12n+1,12n+4)(12n+2,12n+3)(12n+5,12n+6),\\ \beta =(2,8)(3,7)(4,6)\ldots (12n+3,12n+5)(12n+2,12n+6). \end{array}$$Then $G=\langle \alpha ,\beta \rangle \leq A=Aut\left(B_{12n+6}\right)$. Since every automorphism that fixes point 3 also fixes the points {7,26,27,33,…,12n−8,12n−2}, the orbit-stabilizer property implies that $\left|A|=\right|3^{A}\left|\right|A_{3}|=2\times 2$. Therefore, $A≅Z_{2}\times Z_{2}.$ The graph $B_{12n+6}$ has n+1 layers. The orbits of the first and last layers are given by
$$\begin{array}{c} \{1,5\},\{2,4,6,8\},\{3,7\},\{12,13,19,20\},\{12n+1,12n+4\},\\ \{12n+2,12n+3,12n+5,12n+6\}. \end{array}$$Moreover, the vertices of the *i*-th layer (2≤i≤n) of $B_{12n+6}$ that have the same color in Figure 10 are in the same orbit. This means that the graph $B_{12n+6}$ possesses 2n+1 orbits of size 2 and 2n+1 orbits of size 4. Thus,
(33)$$\begin{array}{ccc} I_{a}\left(B_{12n+6}\right) & = & \frac{2(2n+1)}{12n+6}\log\left(\frac{12n+6}{2}\right)+\frac{4(2n+1)}{12n+6}\log\left(\frac{12n+6}{4}\right) \end{array}$$
(34)$$\begin{array}{cc} = & \frac{2n+1}{6n+3}\left(\log(12n+6)-1\right)+\frac{4n+2}{6n+3}\left(\log(12n+6)-2\right) \end{array}$$
(35)$$\begin{array}{cc} = & \log\left(12n+6\right)-\frac{10n+5}{6n+3}. \end{array}$$  □

**Theorem** **8.**
*Suppose $B_{12n+6}$ is the fullerene graph with n≥6. Then*
(36)$$H\left(B_{12n+6}\right)=\log\left(12n+6\right)-\frac{1}{12n+6}\left(6(2n-8)(1+\log3)+90\right).$$


**Proof.** In Figure 10, the sets of Hosoya-equivalent vertices in layers 1, 2, and 3 are precisely the orbits of the automorphism group. For i∈{4,5,6}, consider the *i*-th layer of fullerene $B_{12n+6}$. The vertices labeled 2 and 4 form two blocks of the Hosoya partition. The vertices labeled 1 and 3 form two additional blocks. In the layers *i*, (7≤i≤n), the vertices labeled by 2 and 4 form two blocks, and the vertices labeled by 1 and 3 form an additional two blocks. Finally, the vertices of the last layer are all *H*-equivalent. Hence, the Hosoya partition of this graph consists of nine blocks of size 2, nine of size 4, and 2n−8 of size 6. Thus, we have
$$\begin{array}{ccc} H(B_{12n+6}) & = & \frac{18}{12n+6}\log\left(\frac{12n+6}{2}\right)+\frac{36}{12n+6}\log\left(\frac{12n+6}{4}\right)\\ & + & \frac{6(2n-8)}{12n+6}\log\left(\frac{12n+6}{6}\right)=\frac{3}{2n+1}\left(\log(12n+6)-1\right)\\ & + & \frac{6}{2n+1}\left(\log(12n+6)-2\right) \end{array}$$
(37)$$\begin{array}{cc} + & \frac{2n-8}{2n+1}\log\left((12n+6)-\log6\right) \end{array}$$
(38)$$\begin{array}{cc} = & \log\left(12n+6\right)-\frac{1}{2n+1}\left((2n-8)(1+\log3)+15\right). \end{array}$$  □

**Theorem** **9.**
*If $c_{i}$’s are equal in Equation (Equation 5), then the entropy of fullerene $B_{12n+6}\left(n\geq 12\right)$ is*
(39)$$\begin{array}{ccc} If_{\sigma }\left(B_{12n+6}\right) & = & \log(18n^{2}+18n+8) \end{array}$$
(40)$$\begin{array}{cc} - & \frac{1}{9/2n^{2}+9/2+2}\left((4n+2)\log(2n+1)+5n\log(2n)+3A\right), \end{array}$$
*where $A=\sum_{i=1}^{n-1}\left(2n-i\right)\log\left(2n-i\right)$.*


**Proof.** There exist n+1 types of vertices of fullerene graphs $B_{12n+6}$ whose eccentricity sequence is
(41)$$\begin{array}{c} \left\{{\left(2n-i\right)}^{12}\left(1\leq i\leq n-1\right),{\left(2n\right)}^{10},{\left(2n+1\right)}^{8}\right\}. \end{array}$$There exist 8 vertices with eccentricity 2n+1, 10 vertices with eccentricity 2n, and 12 vertices with eccentricity 2n−i(1≤i≤n−1). We conclude that
$$\begin{array}{ccc} If_{\sigma }\left(B_{12n+6}\right) & = & \log\left(8\left(2n+1\right)+10\left(2n\right)+12\sum_{i=1}^{n-1}\left(2n-i\right)\right)\\ & - & \frac{1}{32n+7+12\sum_{i=1}^{n-1}\left(2n-i\right)}(8\left(2n+1\right)\log\left(2n+1\right) \end{array}$$
(42)$$\begin{array}{cc} + & 10(2n)\log(2n)+12A)) \end{array}$$
$$\begin{array}{cc} = & \log(18n^{2}+18n+8)\\ - & \frac{1}{9/2n^{2}+9/2n+2}(\left(4n+2\right)\log\left(2n+1\right) \end{array}$$
(43)$$\begin{array}{cc} + & 5n\log(2n)+3A)). \end{array}$$  □

**Theorem** **10.**
*The radial entropy of fullerene $B_{12n+6}\left(n\geq 6\right)$ is*
(44)$$H_{ecc}\left(B_{12n+6}\right)=\log\left(12n+6\right)-\frac{1}{12n+6}\left(12(n-1)\log3+10\log5+24n+10\right).$$


**Proof.** By using Equation (Equation 41), we infer
$$\begin{array}{c} H_{ecc}\left(B_{12n+6}\right)=\frac{12(n-1)}{12n+6}\log\left(\frac{12n+6}{12}\right)+\frac{10}{12n+6}\log\left(\frac{12n+6}{10}\right) \end{array}$$
(45)$$\begin{array}{c} +\frac{8}{12n+6}\log\left(\frac{12n+6}{8}\right) \end{array}$$
(46)$$\begin{array}{c} =\log\left(12n+6\right)-\frac{1}{12n+6}\left(12(n-1)\log12+10\log10+8\log8\right) \end{array}$$
(47)$$\begin{array}{c} =\log\left(12n+6\right)-\frac{1}{12n+6}\left(12(n-1)\log3+10\log5+24n+10\right). \end{array}$$  □

**Theorem** **11.**
*The degree-based entropy $D(B_{12n+6})$ is*
(48)$$D\left(B_{12n+6}\right)=\log\left(36n+18\right)-\frac{(36n+18)\log3}{36n+18}=\log\left(12n+6\right).$$


### Correlation Analysis

In Figure 11 and Figure 12, the values of five entropies (introduced in this paper) are compared for 80 fullerene graphs contained in $A_{12n+4}$ and $B_{12n+6}$. Here, the X-axis denotes the values of *n* and the Y-axis denotes the the values of graph entropies. As a result, one can see that the correlation between degree-based entropy and eccentric-entropy is approximately equal to one.

The Pearson correlations between the entropies for the fullerenes $A_{12n+4}$ and $B_{12n+6}$ can be found in Figure 13 and Figure 14.

The adjacency energy of *G* is a graph invariant that was introduced by Gutman [39]. It is defined as
(49)$$\begin{array}{c} E\left(G\right)=\sum_{i=1}^{n}\left|\lambda_{i}\right|, \end{array}$$
where $\lambda_{i}$s are the eigenvalues of *G*. In this paper, we computed the energy of a graph and five types of entropies for $A_{12n+4}$ (11≤n≤20) fullerene graphs (see Table 2). These results reveal that the correlation between graph energy and any type of entropy applied to the class of $A_{12n+4}$ fullerenes is greater than 0.99 (see Table 3). This means that they capture almost the same kind of structural information. Finally, we are able to approximate the graph energies of fullerenes by these entropies.

## 5. Summary and Conclusions

In this paper, we have examined several known graph entropy measures on fullerene graphs. In particular, we explored the topological information content of a graph $I_{a}\left(G\right)$, a degree-based entropy measure, the eccentric-entropy $If_{\sigma }\left(G\right)$, the Hosoya entropy H(G), and finally, the radial centric information entropy $H_{ecc}$. Our results are twofold. First, we obtained concrete expressions for the graph entropy measures on the defined classes of fullerenes. These results can be useful when applying the measures on the fullerenes for practical applications. Second, we generated numerical results to examine the correlations between the measures. We found that almost all measures are highly correlated. This means that it might be sufficient to use only one measure to quantify the structural properties of fullerenes. On the one hand, this could be interpreted as a negative result in that it might not be worthwhile to apply many measures that seem to be different since they rely on quite different graph invariants. However, it turns out that they capture almost the same kind of structural information, as measured by the Pearson correlation coefficient. On the other hand, this fact could be used to approximate other measures that are difficult to determine analytically. In Hückel theory, the total pi-electron energy of a bipartite molecular graph is defined as the formula given by Equation (Equation 49). Our measure of energy correlates well with the observed heats of formation of the corresponding conjugated hydrocarbons, and it is related to other relevant chemical invariants [39,40]. We demonstrated this by using the well-known graph energy [41,42,43,44,45,46,47,48].

In the future, we intend to examine these measures on other classes of graphs and to analyze extremal properties as well as interrelations between the measures.

## Figures and Tables

**Figure 1 entropy-21-00482-f001:**
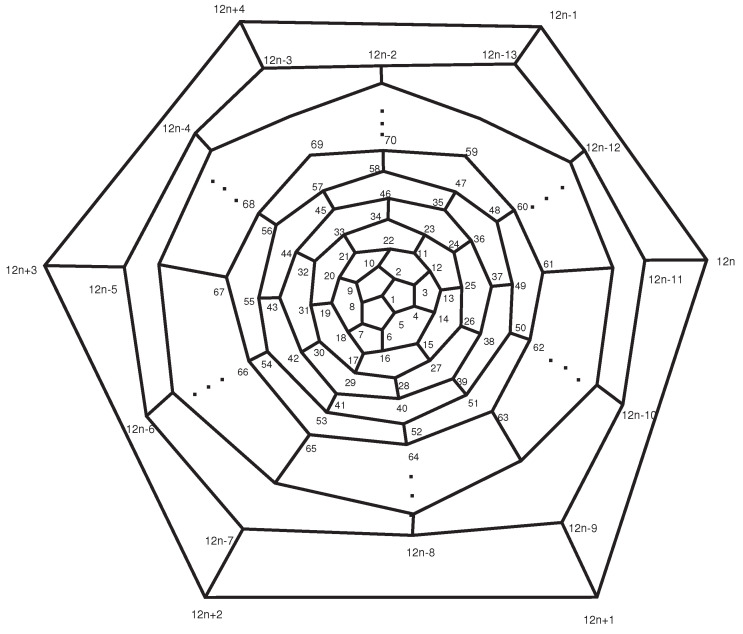
The fullerene $A_{12n+4}$.

**Figure 2 entropy-21-00482-f002:**
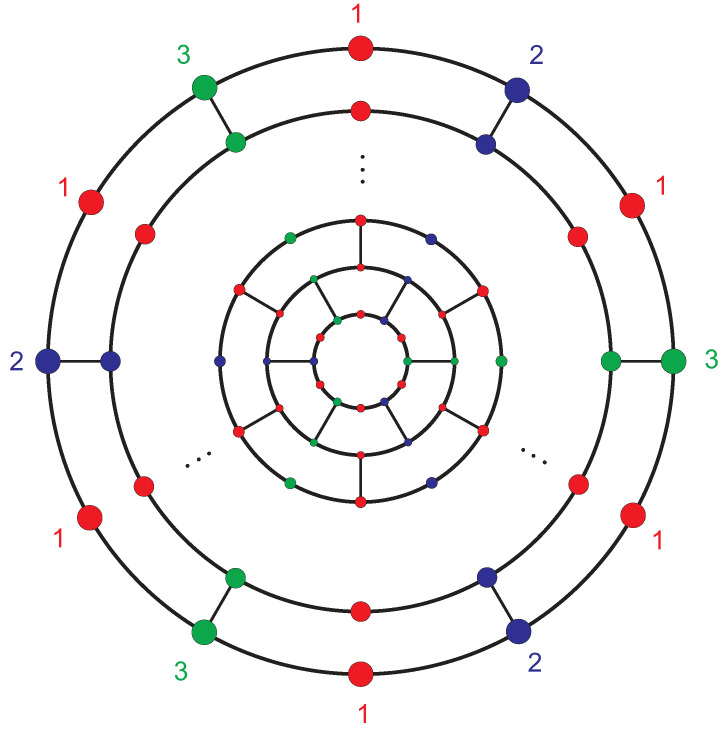
The orbits of the *i*-th layer (2≤i≤n) of the fullerene graph $A_{12n+4}$.

**Figure 3 entropy-21-00482-f003:**
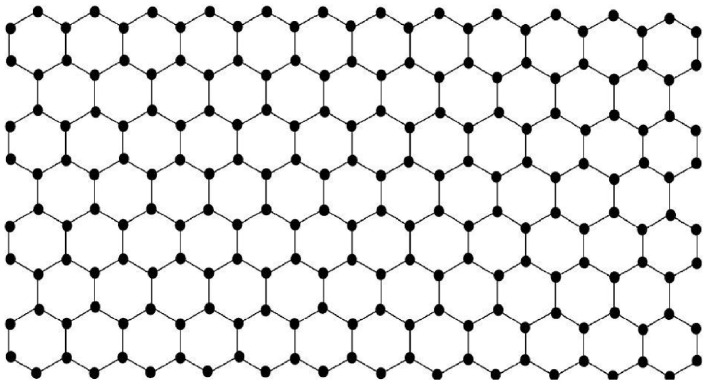
A sheet of hexagons $T_{z}\left[7,12\right]$.

**Figure 4 entropy-21-00482-f004:**
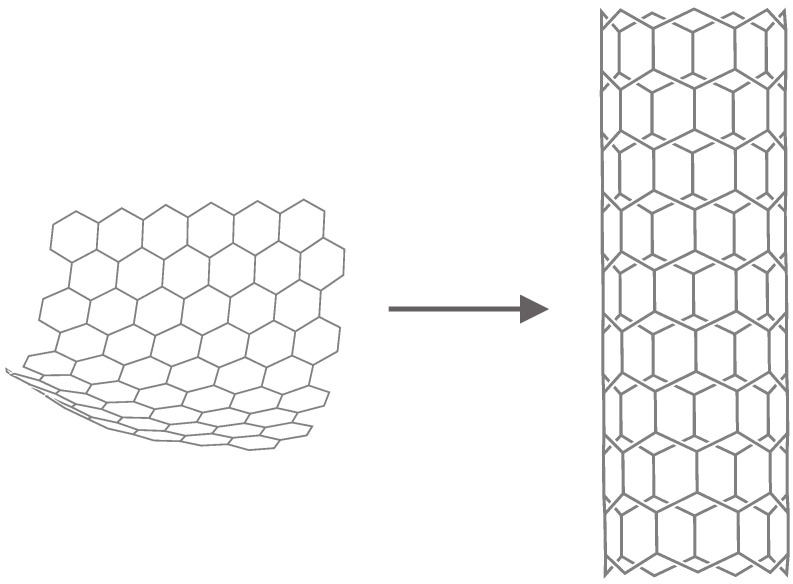
Zig-zag and nanotube.

**Figure 5 entropy-21-00482-f005:**
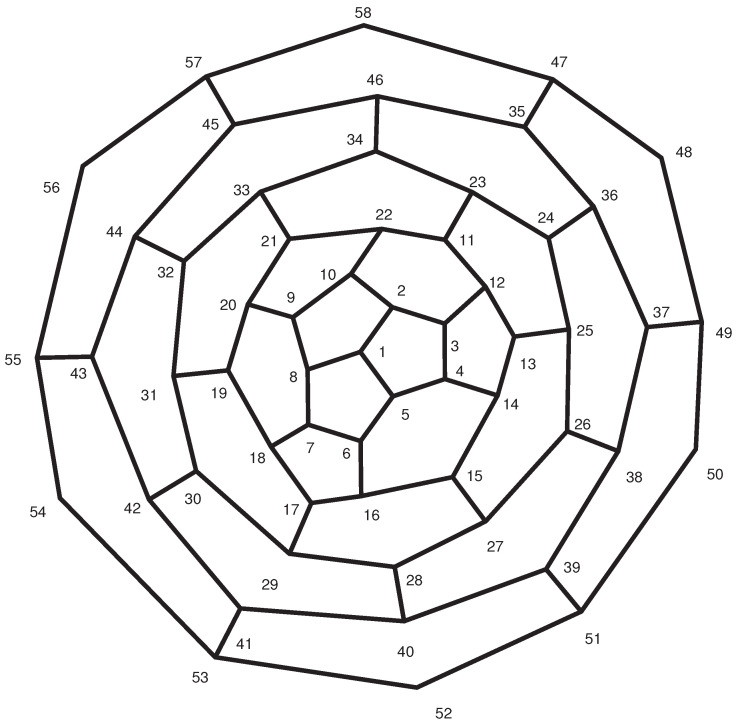
The subgraph $B_{1}$.

**Figure 6 entropy-21-00482-f006:**
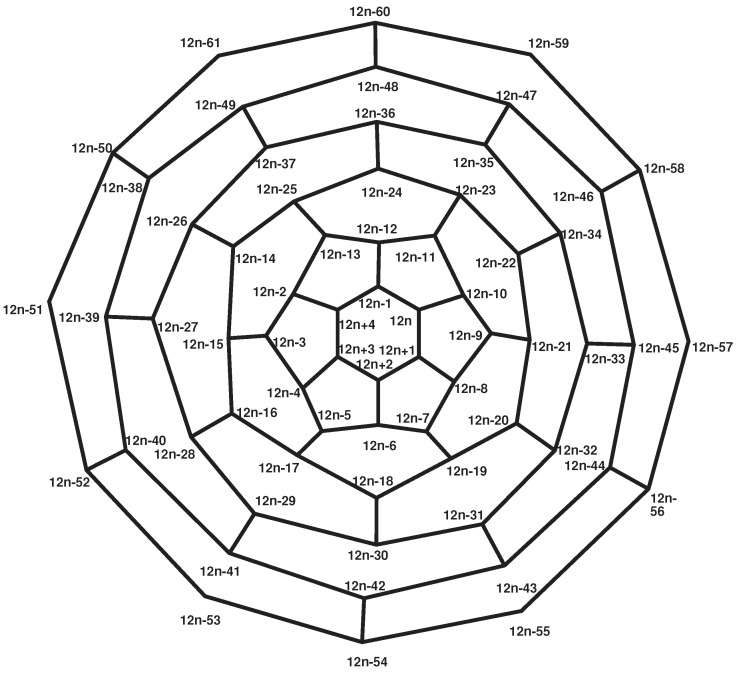
The subgraph $B_{2}$.

**Figure 7 entropy-21-00482-f007:**
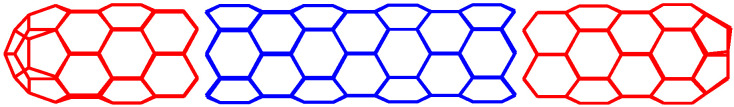
The 3-dimensional structure of fullerene graph $A_{12n+4}$.

**Figure 8 entropy-21-00482-f008:**
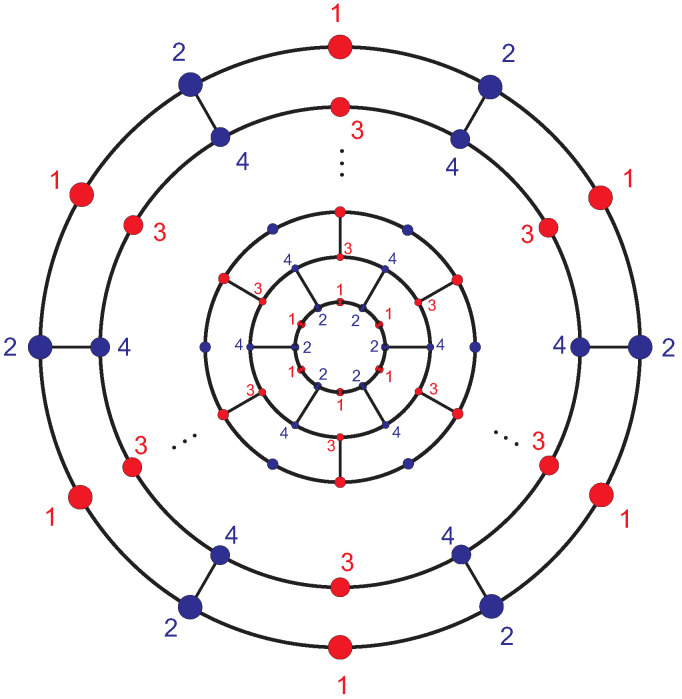
The Hosoya-partitions of $T_{z}\left[6,n-10\right]$.

**Figure 9 entropy-21-00482-f009:**
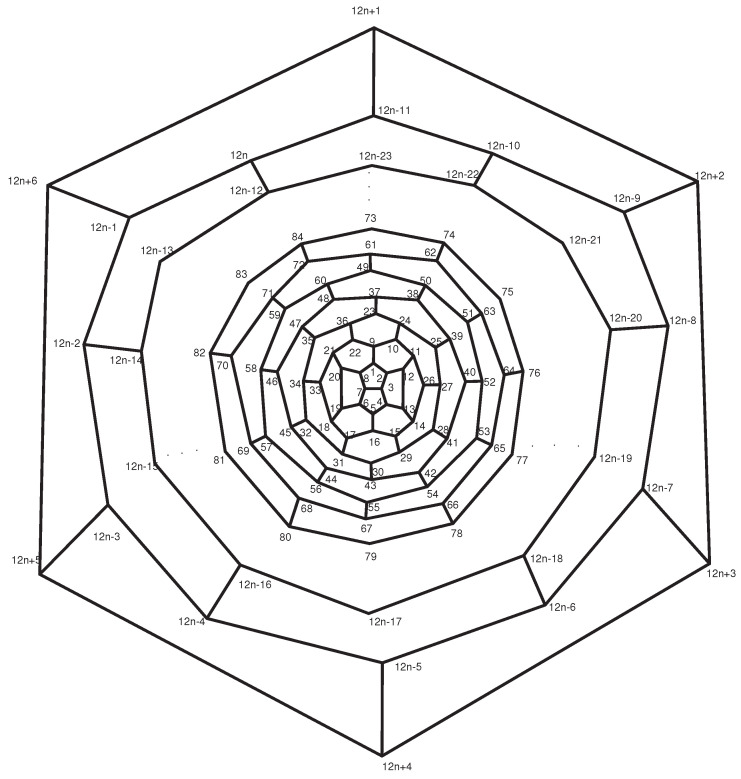
Labeled vertices of the fullerene graph $B_{12n+6}$.

**Figure 10 entropy-21-00482-f010:**
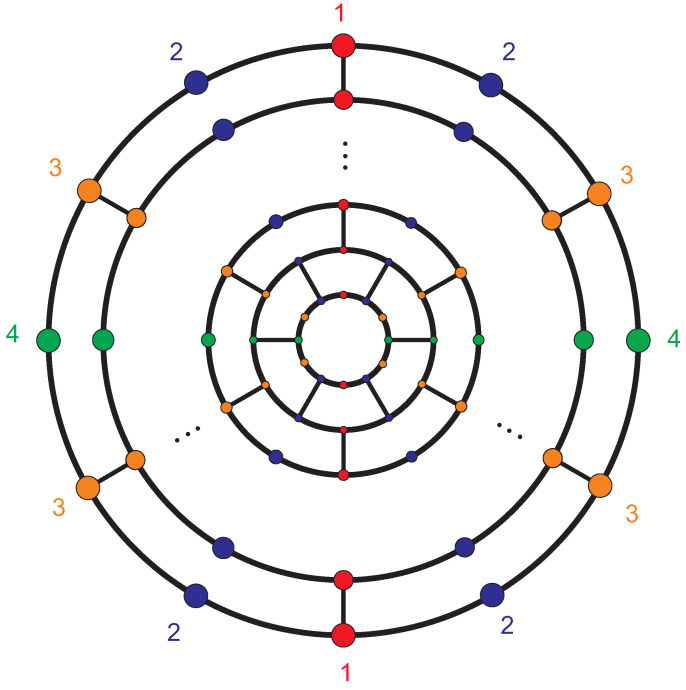
The orbits of the *i*-th layer (2≤i≤n) of the fullerene $B_{12n+6}$.

**Figure 11 entropy-21-00482-f011:**
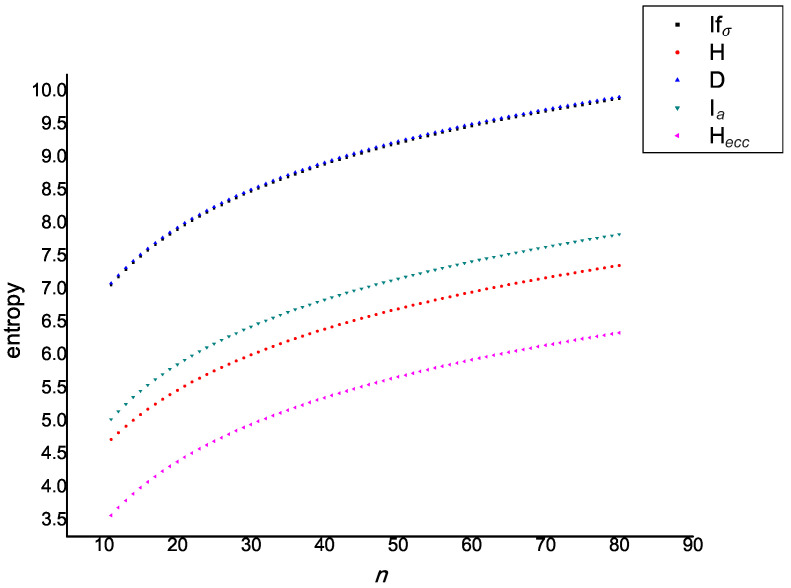
Behavior of graph entropies for the fullerene graph $A_{12n+4}$.

**Figure 12 entropy-21-00482-f012:**
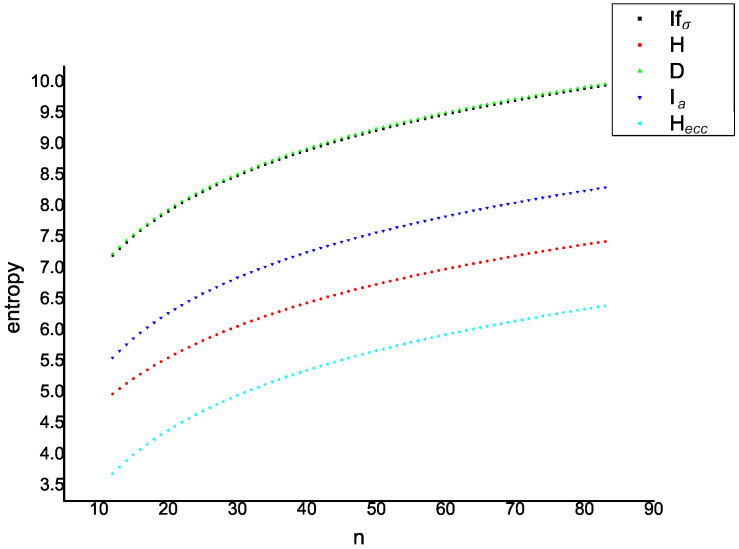
Behavior of graph entropies for the fullerene graph $B_{12n+6}$.

**Figure 13 entropy-21-00482-f013:**
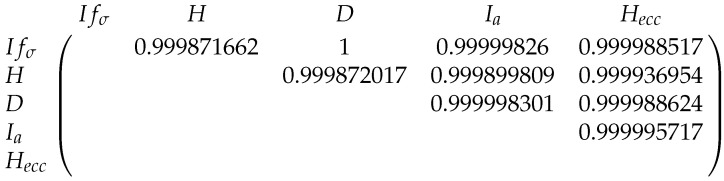
The correlations between five graph entropies for $A_{12n+4}$.

**Figure 14 entropy-21-00482-f014:**
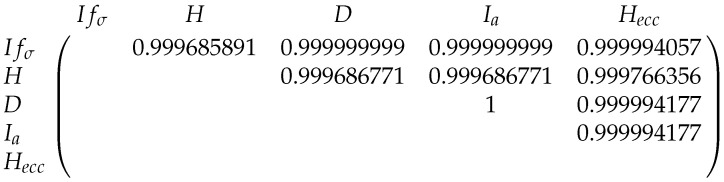
The correlations between five entropies for $B_{12n+6}$.

**Table 1 entropy-21-00482-t001:** The *H*-partition and eccentricity of fullerene graph.

Partitions	Elements	ecc
$V_{1}$	1	2n+1
$V_{2n+6}$	12n−1,12n,12n+1,12n+2,12n+3,12n+4	
$V_{2}$	2,5,8	2n
$V_{2n+5}$	12n−13,12n−11,12n−912n−7,12n−5,12n−3	
$V_{3}$	3,4,6,7,9,10	2n−1
$V_{2n+4}$	12n−12,12n−10,12n−812n−6,12n−4,12n−2	
$V_{4}$	12,14,16,18,20,22	2n−2
$V_{2n+3}$	12n−25,12n−23,12n−21,12n−19,12n−17,12n−15	
$V_{5}$	11,15,19	2n−3
$V_{6}$	13,17,21	
$V_{2n+2}$	12n−24,12n−22,12n−20,12n−18,12n−16,12n−14	
$V_{7}$	23,27,31	2n−4
$V_{8}$	25,29,33	
$V_{2n+1}$	12n−36,12n−34,12n−32,12n−30,12n−28,12n−26	
$V_{9}$	24,26,28,30,32,34	2n−5
$V_{2n}$	12n−37,12n−35,12n−33,12n−31,12n−29,12n−27	
$V_{10}$	36,38,40,42,44,46	2n−6
$V_{2n-1}$	12n−49,12n−47,12n−45,12n−43,12n−41,12n−39	
$V_{11}$	35,39,43	2n−7
$V_{12}$	37,41,45	
$V_{2n-2}$	12n−48,12n−46,12n−44,12n−42,12n−40,12n−38	
$V_{13}$	47,51,55	2n−8
$V_{14}$	49,53,57	
$V_{2n-3}$	12n−60,12n−58,12n−56,12n−54,12n−52,12n−50	
$V_{15}$	48,50,52,54,56,58	2n−9
$V_{2n-4}$	12n−61,12n−59,12n−57,12n−55,12n−55,	
	12n−53,12n−51	

**Table 2 entropy-21-00482-t002:** The graph energy and five kinds of entropies applied to $A_{12n+4}$.

*n*	*E*	*D*	${\mathrm{If}}_{\sigma }$	$I_{a}$	*H*	$H_{\mathrm{ecc}}$
11	212.87	7.08	7.06	5.02	4.72	3.57
12	231.73	7.2	7.18	5.14	4.82	3.68
13	250.59	7.32	7.29	5.25	4.92	3.79
14	269.46	7.42	7.39	5.36	5.01	3.89
15	288.32	7.52	7.49	5.45	5.09	3.98
16	307.19	7.61	7.58	5.54	5.18	4.07
17	326.05	7.7	7.67	5.63	5.25	4.15
18	344.91	7.78	7.75	5.71	5.33	4.23
19	363.78	7.85	7.83	5.78	5.4	4.31
20	382.64	7.93	7.9	5.86	5.46	4.38

**Table 3 entropy-21-00482-t003:** The correlation between graph energy and entropies applied to $A_{12n+4}$.

	E,D	$E,{\mathrm{If}}_{\sigma }$	$E,I_{a}$	E,H	$E,H_{\mathrm{ecc}}$
Cor	0.9964006	0.9972326	0.99673	0.9975728	0.9974525

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
