# Peer review of "On Properties of Distance-Based Entropies on Fullerene Graphs"

_entropy, 2019, doi:10.3390/e21050482_

Round 1
Reviewer 1 Report
The manuscript is devoted to the topological analysis of the fullerene/nanotube structures for further computing the entropy-based indices. I found this material interesting, mathematically correct and well presented. The results obtained may be interesting for fullerene science community and lie in the scope of the Entropy journal.
My rare suggestions deal with incomplete mentioning the first and key works on the topological studies of fullerenes. The following theoretical works should be mentioned in the introduction:
1. Fowler, P. W.; Manolopoulos, D. E. An Atlas of Fullerenes, Oxford: Clarendon Press, 1995; pp. 392 – is a handbook for topologist working with fullerenes.
2. Ori, O.; D’Mello, M. A topological study of the structure of the C76 fullerene, Chem. Phys. Lett. 1992, 197, 49–54 – is the first topological study on fullerenes.
Recent papers of the same author stress the actuality of the topological studies:
3. Ori, O.; Cataldo, F.; Graovac, A. Topological ranking of C28 fullerenes reactivity, Fullerene Nanotube Carbon Nanostruct. 2009, 17, 308–323.
4. Sabirov, D.S.; Ori, O.; Laszlo, I.; Graovac, A. Isomers of the C84 fullerene: A theoretical consideration within energetic, structural, and topological approaches, Fullerene Nanotube Carbon Nanostruct. 2018, 26, 100–110 – recent paper demonstrating the link between topological and information-theory indices.
Another weak spot of the manuscript is the absence of its practical application that can be shortly discussed. The authors wrote in the Conclusions ‘These results can be useful when applying the measures on the fullerenes for practical applications’. It is just a declaration that must be clarified. For example, graph measures are at least relevant to the fullerene reactivity (Ori et al. work 3 above). Additionally, entropy based graph measures may relate to non-equilibrium physicochemical processes:
5. Talanov, V. M.; Ivanov, V. V. Structure as the source of information on the chemical organization of substance. Russ. J. Gen. Chem. 2013, 83, 2225–2336.
6. Aleskovskii, V. B. Information as a factor of self-organization and organization of matter. Russ. J. Gen. Chem. 2002, 72, 569–574.
As for fullerenes, there are direct applications of information entropy to rationalizing the processes of fullerene (work 7 below) and endofullerene (work 8) formation:
7. Sabirov, D.S.; Osawa, E. Information entropy of fullerenes, J. Chem. Inf. Model. 2015, 55, 1576–1584.
8. Sabirov, D.S.; Terentyev, A.O.; Sokolov V.I. Activation energies and information entropies of helium penetration through fullerene walls. Insights into the formation of endofullerenes nX@C60/70 (n = 1 and 2) from the information entropy approach, RSC Adv. 2016, 6, 72230–72237.
I think discussing the results of the manuscript in the context of the mentioned works will clarify the potential of the presented entropy-based graph measures for further fullerene studies.
Author Response
Dear Reviewer,
I added the references and explained in the text more about them.
Reviewer 2 Report
The manuscript topic is on the crossing of two areas of intensive interest - one theoretical (the Shannon entropy) and one applied (fullerenes). The study is well planned and performed. The only recommendation to this high quality and well written manuscript is to go beyond the 1:1 ratio between self- and other aithors citations, paying more attention to closely related publications in these two crossing fields (One such example could be the paper of Sabirov and Osawa "Information Entropy of Fullerenes", JCIM 35, 1576-1584, 2015).
Author Response
Dear Reviewer,
I added the reference and explained more.
Reviewer 3 Report
This is a very accurate and detailed work on the application of graph invariants to fullerene molecules. The authors demonstrate the near to unity correlation among proposed indexes that is interpreted as a way to derive more tricky (and difficult to quantify) properties from such graph invariants. This is surely sound but it is worth noting that ANY CORRELATION COMES FROM THE ANALYSED STATISTICAL UNITS, this is the same concept of shape (a set of invariant correlations among otherwise indpendent parameters). This is why each geometrical form is defined in terms of peculiar relations among geometrical descriptors like area, perimeter, edge length...these relations are specific for each form (thus a square has different relations among its descriptors than triangles or pentagons..).
Thus a set of correlation identifies a specific form (not necessarily regular see Heckman, C. A. (1990). Geometrical constraints on the shape of cultured cells. Cytometry: The Journal of the International Society for Analytical Cytology, 11(7), 771-783.). All fullerenes have basically the same form and the only order parameter is n (the number of atoms, exactly as aliphatic hydrocarbons in which all the chemico-physical properties scale among them and with n), thus the results the authors derive are expected, what could be very interesting is to register how the different descriptors are sensitive to 'errors' i.e. to distortions of the basic ideal structures (this in real world can happen for a number of reasons like impurities or defect in nanotube synthesis, functionalization..). If the authors will insert 'defects' (e.g. varying mutual distances between nodes by adding noise..) in their structure, the different sensitivity to these defects of the proposed descriptors could be a useful information for practical applications.

Author Response
Dear Reviewer,
You are right. The information we can get from the result corr(x,y) is nearly 1 is weak. That means they capture (almost) the same kind of information